# First approach to the population structure of *Mycobacterium tuberculosis* complex in the indigenous population in Puerto Nariño-Amazonas, Colombia

**Alejandro Vega Marín**[1], **Nalin Rastogi**[2], **David Couvin**[2], **Viviana Mape**[1], **Martha Isabel Murcia**[1] *

**1** MICOBAC-UN, Departamento de Microbiología, Facultad de Medicina, Universidad Nacional de Colombia, Bogotá, Colombia, **2** WHO Supranational TB Reference Laboratory, Unité de la Tuberculose et des Mycobactéries, Institut Pasteur de la Guadeloupe, Abymes, Guadeloupe, France

* mimurciaa@unal.edu.co

**Data Availability Statement:** All relevant data are within the manuscript and its Supporting Information files.

## Abstract

### Introduction

Tuberculosis affects vulnerable groups to a greater degree, indigenous population among them.

### Objective

To determine molecular epidemiology of clinical isolates of *Mycobacterium tuberculosis* circulating in an indigenous population through Spoligotyping and 24-loci MIRU-VNTR.

### Methodology

A descriptive cross-sectional study was conducted in 23 indigenous communities of Puerto Nariño-Amazonas, Colombia. Recovered clinical isolates were genotyped. For genotyping analyzes global SITVIT2 database and the MIRU-VNTRplus web portal were used.

### Results

74 clinical isolates were recovered. Genotyping of clinical isolates by spoligotyping determined 5 different genotypes, all of them belonged to Euro-American lineage. By MIRU-VNTR typing, a total of 14 different genotypes were recorded. Furthermore, polyclonal infection was found in two patients from the same community. The combination of the two methodologies determined the presence of 19 genotypes, 8 formed clusters with 63 clinical isolates in total. Based on epidemiological information, it was possible to establish a potential chain of active transmission in 10/63 (15.9%) patients.

### Conclusions

High genomic homogeneity was determined in the indigenous population suggesting possible chains of active transmission. The results obtained showed that specific genotypes

**Funding:** The research was funded by Fondo de Ciencia, Tecnología e Innovación del Sistema General de Regalías BPIN 2013000100230, agreement 000063 between Gobernación de Amazonas and Universidad Nacional de Colombia. The TB phylogenetic analysis at Institut Pasteur de la Guadeloupe benefitted from a FEDER grant financed by the European Union and Guadeloupe Region (grant number 2015-FED-192). The funders had no role in study design, data collection and analysis, decision to publish, or preparation of the manuscript.

**Competing interests:** The authors have declared that no competing interests exist.

circulating among the indigenous population of Colombia are significantly different from those found in the general population.

## Introduction

Tuberculosis (TB) remains a global emergency. According to the World Health Organization (WHO), TB is the leading cause of death due to a single infectious agent, and it is among the 10 leading causes of death in the world [1]. In 2016, WHO estimated there were 10.4 million new cases of TB worldwide for a global incidence of 140 cases per 100,000 inhabitants [2].

For a national incidence of 25.7 cases per 100,000 inhabitants in Colombia in 2016 (13,871 cases of TB, 12,439 of which were new cases), a total of 840 cases were notified in the indigenous population. The Amazonas department reported 128 new TB cases, corresponding to an incidence rate of 163.4 cases per 100,000 inhabitants, which is the highest in the country [3]. Indeed, the indigenous population has been defined by national and international health agencies as at high risk for the transmission of tuberculosis, due to environmental conditions, sanitary hygiene, overcrowding, malnutrition, difficulties of geographic access to communities; and poor or uncomfortable environment, according to their beliefs, in health facilities for indigenous populations [4].

Molecular epidemiology is a discipline that combines the techniques of classical epidemiology with molecular techniques, in order to determine the transmission dynamics of infectious diseases, including TB. The method most commonly used previously for the genotyping of *Mycobacterium tuberculosis* complex (MTBC) was IS*6110*-restriction fragment length polymorphism [5]. Being particularly cumbersome, this technique was gradually replaced by PCR-based spoligotyping and 24-loci MIRU-VNTR typing [6, 7]. The association of these 2 methods is particularly helpful since spoligotyping based on the polymorphism of the DR locus [8] simultaneously classifies individual MTBC isolates in broad genotypic families, and helps predict lineages from genotypic patterns using an international database [9, 10]. Drawing and analysing dendrograms further allows to determine phylogenetic and evolutionary relationships between MTBC isolates circulating in a given geographic region [11–13]. Last but not least, the addition of 24-loci MIRU-VNTR, method based on the variation in the number of repetitions of the MIRUs loci analyzed [14], unequivocally allows the grouping of clinical isolates with the same genotypic pattern in clusters indicating potential transmission events. Conversely, unique patterns may underline potential events of endogenous reactivation [11, 14]. The combination of these 2 molecular techniques allows today to identify the genotypes of *M. tuberculosis* circulating both globally and locally, which is important to understand the dynamics of disease transmission [10]. Although not used in the present investigation, whole genome sequencing offers today a robust phylogenetic framework for classification of strains as demonstrated by Walker et al., who sequenced 390 isolates of 254 patients and showed that 5 or more single nucleotide polymorphisms (SNPs) might separate MIRU-based clusters [15]. The epidemiological significance of determining the genotypes and lineages lies in the fact that according to studies in animal models, certain lineages and sub-lineages caused greater progression to active disease; were potentially more virulent, and could be implicated in outbreaks of drug-resistant TB [16].

The objective of the present study was to genotypically characterize MTBC clinical isolates obtained from an indigenous population in Puerto Nariño—Amazonas (Colombia), using the spoligotyping and 24-loci MIRU-VNTR methods, and compare the genotypes observed with those circulating in general (non-indigenous) population of Colombia.

## Materials and methods

### Clinical isolates

The study included 74 clinical isolates belonging to the MTBC obtained from patients diagnosed with pulmonary TB by smear microscopy and/or culture, who were found by house-to-house medical consultations within an entire population of 23 indigenous communities in Puerto Nariño—Amazonas (Colombia). The region is at southwest of the Amazonian Trapeze (03˚ 54' and 03˚ 12' south latitude and 70˚ 17' and 70˚ 42' western longitude). Puerto Nariño is a township with difficult geographical access. The only access is by river through the Amazon River. 62 patients were diagnosed by primary medical consultation, while 12 were diagnosed during contact-tracing. A standardized questionnaire was used by the physician during medical consultation to obtain demographic information (gender, date of birth, community of residence, ethnicity, level of education), clinical evaluation (presence and length of cough and/or expectoration, characteristics of sputum, temperature, weight, height, blood pressure, respiratory distress, weight loss, hemoptysis) and risk factors (prior contact with patients with TB, prior diagnosis of TB). The strains were isolated from March 15th to October 31st 2016 and concerned a population documented as indigenous (although 4/74 patients acknowledged themselves as non-indigenous after inclusion). The sputum samples were analyzed by bacilloscopy in Puerto Nariño and subsequently sent for culture each time they were collected to Leticia, Amazonas. Positive cultures were kept refrigerated until moved to Bogotá for molecular analysis.

### Ethical approval

The study was approved by the ethics committee of the School of Medicine, Universidad Nacional de Colombia and by Association of Indigenous Authorities, ATICOYA. Individuals who agreed to participate voluntarily in the study signed the informed consent after prior explanation of the project and clarification of doubts. For minors (children under 18 years old), the consent was signed by one of the parents or legal representatives.

### Clinical isolates and DNA identification

The 74 clinical isolates in this study were recovered from spontaneous or induced sputum samples by culture, in two tubes of Löwenstein-Jensen medium (LJ) and one tube of BAC-TEC™ MGIT medium. The identification was made using the BD MGIT™ TBc Identification Test (Becton Dickinson, Sparks MD). In case of doubtful and/or negative results, the GeneXpert® MTB/RIF (Cepheid, Sunnyvale, CA, USA) and GenoType®MTBDR plus 2.0 (Hain Lifescience, Germany) tests were used, that in addition to confirming the presence of *Mycobacterium tuberculosis* complex DNA, also helped determine whether the isolate was resistant to rifampin (RR), or both to rifampin and isoniazid (MDR), respectively.

### DNA extraction

DNA extraction was carried out from cultures obtained in Löwenstein Jensen medium using the commercial kit GenoLyse® (Hain Lifescience, Germany) according to the manufacturer's instructions.

### Genotyping and molecular data analysis

DNA extracts from grown cultures were used for genotyping using spoligotyping and 24-loci MIRU-VNTR typing. Spoligotyping was performed as reported previously [8], using a commercial membrane (Mapmygenome India limited). Interpretation was performed manually

according to the absence or presence of each of the 43 spacers; it was reported as 0 (absence of hybridization) or 1 (presence of hybridization).

24-loci MIRU-VNTR typing was performed as described previously [17]. Briefly, the DNA extracts were subjected to individual PCR, and the PCR products were subjected to electrophoresis in 2% agarose gel stained with SYBR Safe (Invitrogen) for 180 minutes at 120 V. The visualization was performed by the photo-documenting equipment of ChemiGenius gels (Syngene) and GeneSnap® software. Molecular weight determination was performed with GeneTools® software version 6.07. The number of repetitions was determined using the conversion table provided by Supply et al [17].

Database analysis was performed using the SITVIT2 database of Pasteur Institute of Guadeloupe [10] (publicly available at http://www.pasteur-guadeloupe.fr:8081/SITVIT2/), which is an updated version of the SpolDB4 and SITVITWEB [18] databases; and the MIRU-VNTR-plus application [19, 20] available at https://www.miru-vntrplus.org/MIRU/index.faces. The MIRU-VNTRplus portal was also used for the construction of dendrograms by using the UPGMA (Unweighted Pair Group Method with Arithmetic Average) method according to the results of spoligotyping and MIRU-VNTR, obtaining by this way the number of formed clusters. The SITVIT2 database was used to assign genotypes and to compare them with those reported in other regions worldwide. The genotyping results were combined with epidemiological data obtained from the questionnaires to indicate whether the patients grouped in the same cluster were potentially involved in a transmission chain. Finally, BioNumerics software version 6.6 (Applied Maths, Sint-Martens-Latem, Belgium; http://www.applied-maths.com/bionumerics) and SpolTools software [12, 13] (available at http://spoltools.emi.unsw.edu.au/), were used to draw minimum spanning trees (MSTs) and a spoligoforest (displayed as an hierarchical layout) respectively, in order to highlight potential relationships between clinical isolates.

## Results

### Study population

Puerto Nariño is a dispersed rural municipality of Amazonas Department, Colombia, located at extreme southeast of Colombia, bordering Peru, on the shore of the Amazon River and formed by 23 indigenous communities. The indigenous communities are isolated from each other and the only means of transport is by river. At the time of the medical consultation, demographic characteristics of the total population of 6310 inhabitants vs. 74 TB patients were summarized (Table 1). The distribution of the total population (6310) was 52% male vs. 48% female, additionally the age range was from 0 to 96.5 years, with an average of 17.5 years. On the other hand, the age range of the patients (this study) was from 1–82 years (average of 30 years). The age group 5–9 years with 12 patients (16.2%) was the main group among diagnosed cases, and almost half of the cases (47.3%) were in the range of 1–19 years. Regarding the ethnicity of the TB patients, 37 patients (50%) belonged to the Ticuna ethnic group, 14 patients (18.9%) belonged to the Cocama ethnic group and 19 individuals (25.7%) belonged to the Yagua ethnic group. The 4 non-indigenous patients represented 5.4%. 73/74 clinical isolates were sensitive to rifampicin and isoniazid, while a single clinical isolate was classified as MDR-TB, and belonged to SIT42/LAM9.

### Genotyping analysis

By spoligotyping it was confirmed that 100% of isolates corresponded to the species *M. tuberculosis* and belonged to the Euro-American lineage (Table 2). 73/74 (98.65%) isolates shared 4 identical genotypes that were grouped into 4 clusters. 1/74 (1.35%) clinical isolate was not

**Table 1. Demographic characteristics of the population and patients with clinical isolates of *M. tuberculosis* in this study.**

| Variables | | Total Population | | Patients with TB isolates [*] | |
|---|---|---|---|---|---|
| | | Frequency | Percentage | Frequency | Percentage |
| **Sex** | Male | 3282 | 52% | 45 | 61% |
| | Female | 3028 | 48% | 29 | 39% |
| **Indigenous people** | Ticuna | 4432 | 70,2% | 37 | 50% |
| | Cocama | 830 | 13,2% | 14 | 18.9% |
| | Yagua | 465 | 7,4% | 19 | 25.7% |
| | Other | 110 | 1,7% | 0 | 0% |
| **Age** | 0–4 | 983 | 15,6% | 8 | 10.8% |
| | 5–9 | 905 | 14,3% | 12 | 16.2% |
| | 10–14 | 815 | 12,9% | 9 | 12.2% |
| | 15–19 | 675 | 10,7% | 6 | 8.1% |
| | 20–24 | 426 | 6,8% | 3 | 4% |
| | 25–29 | 445 | 7,1% | 2 | 2.7% |
| | 30–34 | 403 | 6,4% | 4 | 5.4% |
| | 35–39 | 330 | 5,2% | 3 | 4% |
| | 40–44 | 309 | 4,9% | 4 | 5.4% |
| | 45–49 | 245 | 3,9% | 5 | 6.8% |
| | 50–54 | 179 | 2,8% | 7 | 9.5% |
| | 55–59 | 178 | 2,8% | 3 | 4% |
| | ≥ 60 | 408 | 6,5% | 8 | 10.8% |
| | Unknown [**] | 9 | 0,1% | - | - |
| **Indigenous communities** | Puerto Nariño | 1718 | 27,2% | 13 | 17.6% |
| | San Martín de Amacayacu | 538 | 8,5% | 1 | 1.4% |
| | Puerto Esperanza | 452 | 7,2% | 3 | 4.1% |
| | San Francisco | 409 | 6,5% | 9 | 12.2% |
| | Ticoya | 406 | 6,4% | 7 | 9.5% |
| | Naranjales | 372 | 5,9% | 4 | 5.4% |
| | 20 de Julio | 256 | 4,1% | 4 | 5.4% |
| | 12 de Octubre | 254 | 4,0% | 4 | 5.4% |
| | Atacuary | 243 | 3,9% | 7 | 9.5% |
| | 7 de Agosto | 230 | 3,6% | 4 | 5.4% |
| | San Juan del Socó | 193 | 3,1% | 3 | 4.1% |
| | Tipisca | 191 | 3,0% | 3 | 4.1% |
| | Patrullero | 155 | 2,5% | - | - |
| | Puerto Rico | 155 | 2,5% | 3 | 4.1% |
| | Buhayahuazú | 154 | 2,4% | 6 | 8.1% |
| | Nuevo Paraíso | 122 | 1,9% | - | - |
| | Villa Andrea | 113 | 1,8% | - | - |
| | Palmeras | 98 | 1,6% | - | - |
| | Valencia | 88 | 1,4% | 2 | 2.7% |
| | Santa Teresita | 63 | 1,0% | 1 | 1.4% |
| | Santarem | 57 | 0,9% | - | - |
| | Tarapoto | 37 | 0,6% | - | - |
| | Others | 6 | 0,1% | - | - |

[*] Note that 19/74 (25.7%) patients were diagnosed with TB both by smear microscopy and culture, while 55/74 (74.3%) were diagnosed only by culture.

[**] People who do not remember age or date of birth.

**Table 2. Description and global distribution of predominant SITs representing 2 or more isolates in this study.**

| SIT | Spoligotype Description | Octal code | Number in study (%) | % in study vs. SITVIT database | Lineage | Distribution in countries with ≥3% of a given SIT* | Distribution in regions with ≥3% of a given SIT** |
|---|---|---|---|---|---|---|---|
| 42 |  | 777777607760771 | 2 (2.7) | 0.05 | LAM9 | BR = 15.25, US = 10.32, CO = 7.48, MA = 6.13, IT = 5.69, FR = 4.42, AR = 3.43, PE = 3.22 | South America = 35.32, Southern Europe = 10.65, North America = 10.32, Western Europe = 8.24, North Africa = 7.46, Northern Europe = 4.26, West Indies = 3.69, East Africa = 3.38, Central America = 3.09 |
| 95 |  | 777777607560731 | 37 (50) | 30.95 | LAM6 | BR = 37.30, CO = 30.95, PE = 12.70, US = 6.35, GF = 3.97 | South America = 86.51, North America = 6.35 |
| 392 |  | 777777600160731 | 20 (27.03) | 80.0 | T2*** | CO = 80.00, BR = 16.00, AR = 4.00 | South America = 100.00 |
| 1355 |  | 777777407560731 | 14 (18.92) | 7.04 | LAM | PE = 81.41, CO = 7.04, IT = 5.03, US = 3.52 | South America = 89.45, Southern Europe = 6.53, North America = 3.52 |

\* The 2 letter country codes are according to https://en.wikipedia.org/wiki/ISO_3166-1_alpha-2; countrywide distribution is only shown for SITs with ≥3% of a given SIT as compared to their global distribution in the SITVITEXTEND database.

\*\* Worldwide distribution is reported for regions with more than 3% of a given SIT as compared to their global distribution in the SITVITEXTEND database. The definition of macro-geographical regions and sub-regions (https://unstats.un.org/unsd/methodology/m49/) is according to the United Nations.

\*\*\* Note that SIT392 was relabeled as belonging to LAM according to MIRU-VNTR analysis.

found in the SITVIT2 database, and corresponded to an orphan pattern. 53/74 (71.62%) isolates corresponded to the LAM sublineage and 20/74 (27.03%) to the T sublineage. SITVIT2 database analysis revealed that 2 (2.70%) isolates belonged to SIT42/LAM9, 14 (18.92%) isolates to SIT1355/LAM, 20 (27.03%) isolates to SIT392/T2 and 37 (50.00%) isolates to SIT95/LAM6. SIT392 was probably mislabeled as belonging to the T family due to peculiar absence of spacers (21–29, 33–36, and 40). The spoligotyping pattern designed as an orphan in this study (with a supplementary absent spacer 15) was similar to SIT392. Both spoligotypes could be relabeled as belonging to LAM family (according to MIRU-VNTR analysis).

The 24-loci MIRU-VNTR typing revealed that 66/74 (89.2%) clinical isolates shared 6 identical genotypes and were grouped into 6 clusters (with a size between 2 to 28 patients), and 8/74 isolates (10.8%) presented unique MIRU-VNTR patterns. In addition, a polyclonal infection was detected in 2 patients since they presented 2 bands in 3 different loci (MIRUs 26, 43 and 49) (Fig 1). Molecular analyses performed in conjunction with the two methodologies (spoligotyping and MIRU-VNTR typing) revealed the presence of 19 genotypes. 63/74 (85.1%) clinical isolates formed 8 clusters and 11/74 (14.9%) isolates presented a single genotype. Additionally, in two patients who developed polyclonal infection (detected by both MIRU-VNTR and spoligotyping-MIRU-VNTR typing) exclusive clusters were formed (Fig 1).

Finally, minimun spanning trees (MST) generated with the combination of spoligotyping and MIRU-VNTR patterns (Fig 2), allowed us to observe potential relationships between the 76 genotypes of *M. tuberculosis* found in 74 isolates (since 4 genotypes from 2 patients corresponded to polyclonal infections). One may notice that isolates previously labeled as belonging to T family (relabeled as LAM according to MIRU-VNTR analysis) were well clustered

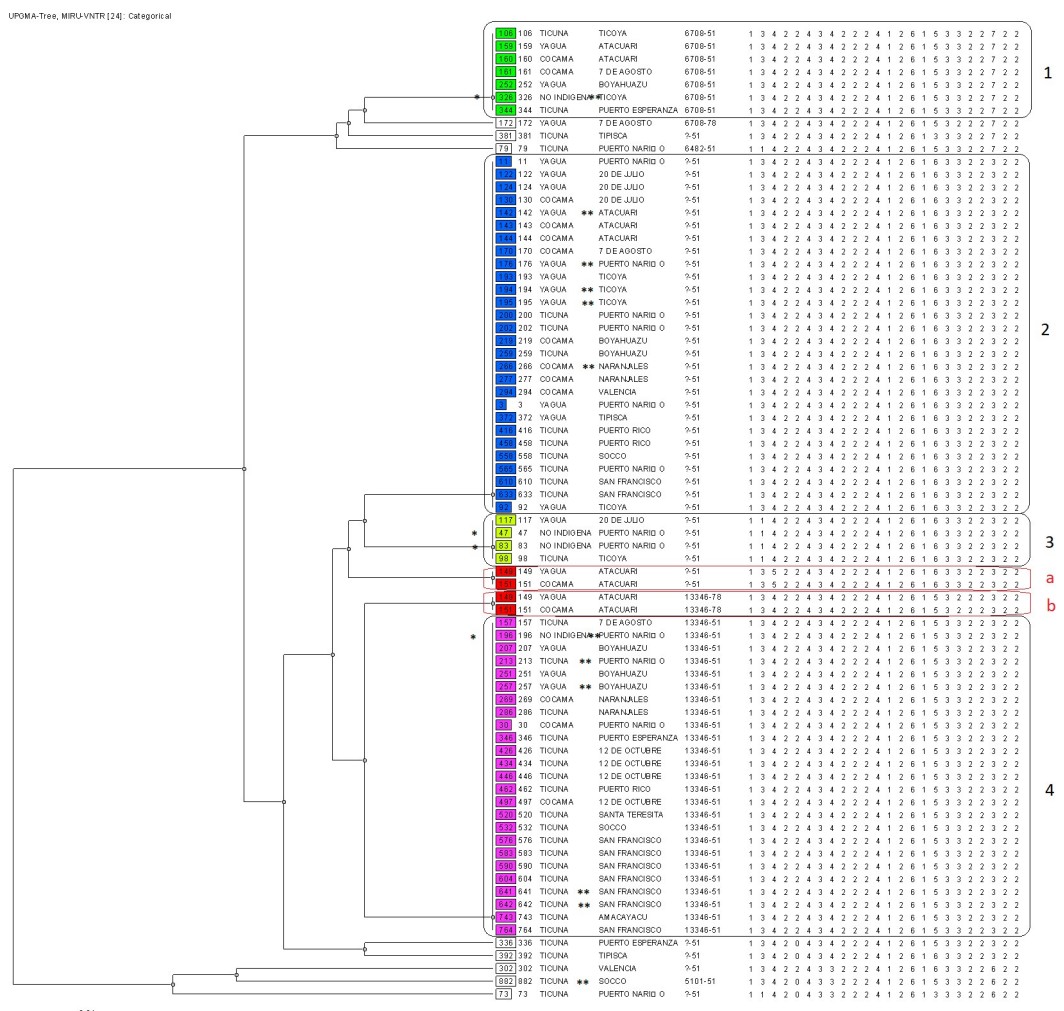

**Fig 1. Dendrogram based on 24-loci MIRU-VNTRs drawn using UPGMA algorithm.** The UPGMA tree indicates the comparison of 74 isolates based on MIRU-VNTR results. From left to right are shown: identification of isolated, ethnic group, community of residence, MLVA MtbC15-9 type and MIRU-VNTR pattern. Numbers indicate the main clusters, and letters "a" and "b" (written in red), indicate the polyclonal infections. One asterisk (*) indicates non-indigenous patients, whereas two asterisks (**) indicate patients found through contact tracing.

together, particularly in the MST based on spoligotyping (Fig 2A) and the MST combining both spoligotyping and 24-loci MIRU-VNTR (Fig 2C).

## Epidemiological analysis of clusters

12/74 patients were found through contact tracing. 6/12 contacts presented the same genotype as the respective index case (Fig 1). The 8 clusters formed by combining both spoligotyping and 24-loci MIRU-VNTR were analyzed including epidemiological links determined through the questionnaires applied in the study population. Cluster 1 consisted of 7 clinical isolates; the patients were distributed in 5 communities, two patients from Ticoya community resided in the same household. Cluster 2 formed by 3 clinical isolates, were distributed in 7 de Agosto and San Francisco communities. Cluster 3 with 21 clinical isolates, was the largest cluster and was distributed in 10 indigenous communities; in San Francisco community lived 5 patients, 3 of whom are brothers residing in the same home; the others patients were in: 12 de Octubre

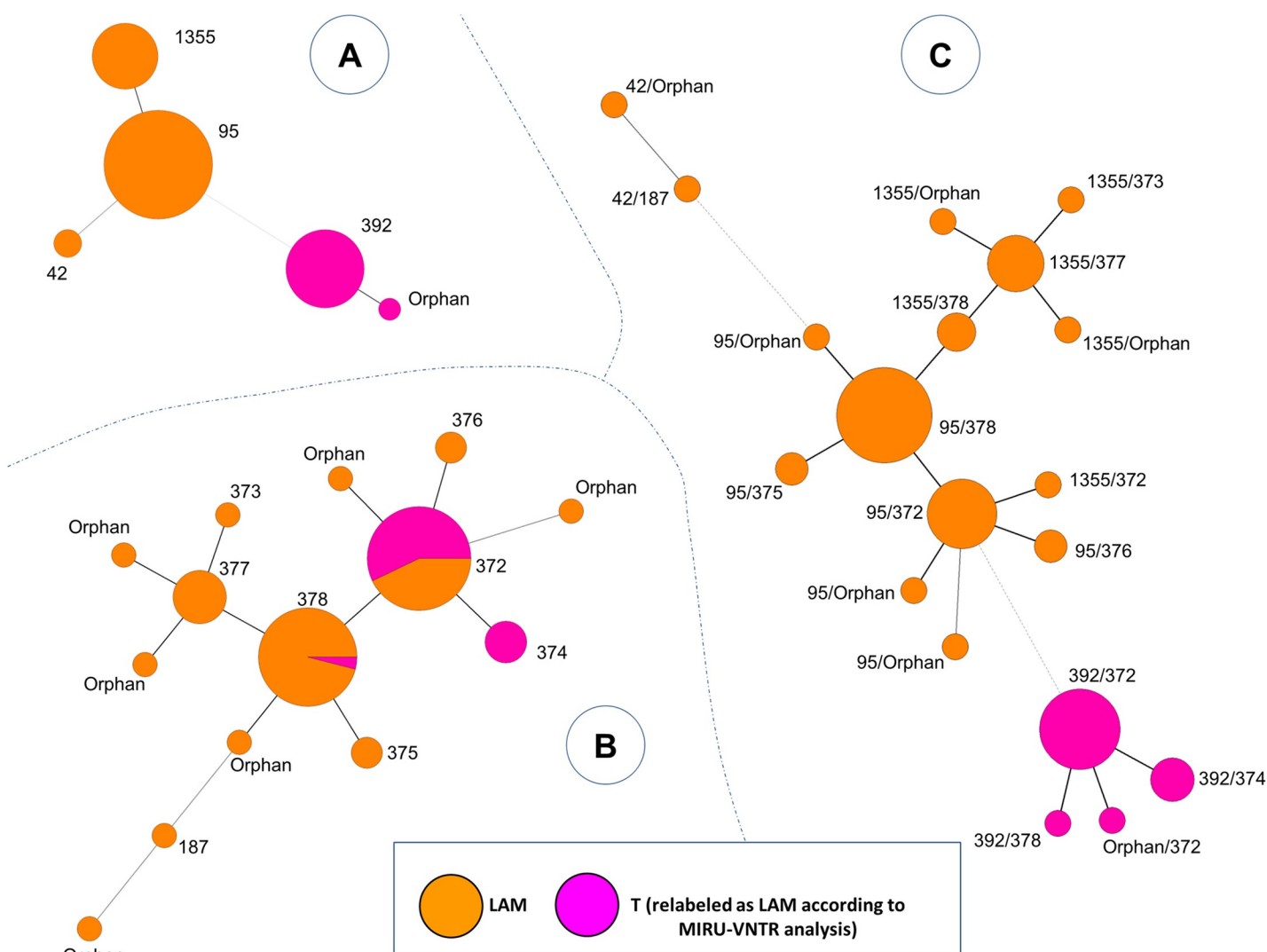

**Fig 2. Minimum spanning tree showing relationships between *M. tuberculosis* isolates.** The tree connects each genotype based on degree of changes required to go from one allele to another. The tree is represented by branches (thicker, thinner or dashed lines representing the degree of difference between patterns) and circles representing each individual pattern (the circle size is proportional to number of isolates for a given pattern). (A) MST constructed with spoligotyping alone. (B) MST constructed with 24-loci MIRU-VNTRs alone. (C) Composite MST with spoligotyping and 24-loci MIRU-VNTRs patterns. MST were constructed on all genotypes (n = 76, including 2 isolates with polyclonal infection).

(n = 4), Boyahuazu (n = 3), Naranjales (n = 2) and Puerto Nariño (n = 2). Cluster 4 consisted of 11 clinical isolates that were distributed in 9 indigenous communities. Cluster 5 consisted of 15 clinical isolates distributed in 7 communities, 3 patients from Ticoya community resided in the same home, while Puerto Nariño community grouped 3 patients, two of whom lived in the same household; 20 de Julio and Atacuari communities grouped 3 patients each. Cluster 6 was formed by 4 clinical isolates and patients were distributed in 3 communities. Finally, clusters "a" and "b" were formed by two patients with polyclonal infection, which were neighbors and had a sentimental relationship (S1–S11 Figs). Unless otherwise indicated, patients didn't share a household but it must be pointed out that indigenous communities interact frequently in social scenarios, therefore, an active chain of transmission cannot be discarded.

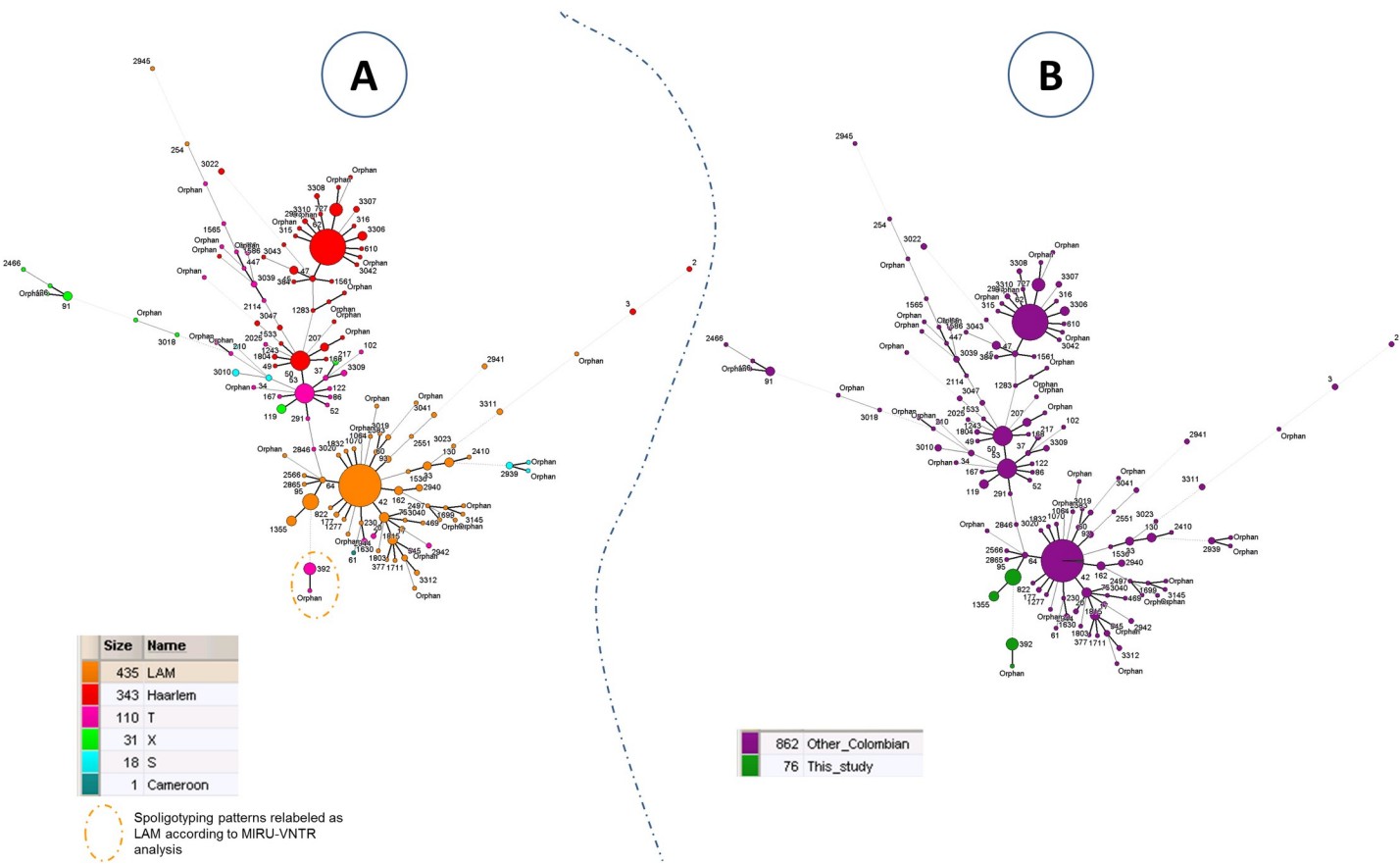

**Fig 3. Minimum spanning tree showing relationships between *M. tuberculosis* isolates in Colombia.** Spoligotypes from this study as compared to other Euro-American spoligotypes from Colombia (n = 938 isolates, including the 74 isolates from the present study that comprised 76 genotypes). (A) MST based on spoligotyping lineages according to SITVIT2; (B) MST based on the origin of patients from this study as compared to patients from other regions of Colombia. Size of nodes or complexity of lines are the same as in Fig 2.

## Comparison with other Euro-American spoligotypes found in Colombia

Finally, a MST (Fig 3) and a spoligoforest (S12 Fig) were drawn to compare spoligotyping patterns from this study as compared to other Euro-Amercian isolates (Cameroon, Haarlem, LAM, S, T, Turkey, X) from Colombia (n = 938 isolates, including the 74 isolates/76 genotypes from the present study; since 4 genotypes from 2 patients corresponded to polyclonal infections). Both trees showed that spoligotypes from Indigenous population (this study) clustered together and were localized in the bottom of the trees. Spoligotypes belonging to the indigenous population were linked to SIT64/LAM6 in the MST (Fig 3) and to SIT60/LAM4 in the spoligoforest (S12 Fig).

## Discussion

The current study allowed to observe the prevailing situation of tuberculosis in indigenous population of Puerto Nariño-Amazonas, Colombia. A total of 6310 persons were subjected to medical consultations, 5837 of whom recognized themselves as indigenous. Among 80 cases diagnosed, 6 were classified as relapses, according to information found in the Tuberculosis Program of the Department of Amazonas, indicating a prevalence of 1267 cases/100 000 inhabitants. The perception of the disease as well as overcrowding have been found as risk

factors for relapse [21]. Furthermore, as shown by Moreno-Martinez et al., inadequate management of the health team negatively influences patients [22]. In the population studied, these risk factors converge because of overcrowding as well as due to socio-demographic characteristics that result in a population with a different perception of the disease [4]. Moreover, a difficult access further prevents adequate monitoring by the health system, leading to an increased risk of relapse. There is no molecular typing information available for previous isolates, so reinfection or reactivation could not be ruled out. The monitoring and strengthening of tuberculosis programs, especially in vulnerable populations is necessary for an adequate management of the disease as well as to know the biological behavior of the MTBC. Limiting solely to the indigenous population (n = 5837), 76 cases were diagnosed, 6 of which were classified as relapses, which corresponds to a prevalence of 1302 cases / 100 000 inhabitants, which is 48 times higher than the prevalence found in the general population of Colombia in 2015 (26.8 cases/ 100 000 inhabitants). The number of cases found differs significantly from those found in previous years. For instance, only 52 cases of TB were reported in 2015 for the entire Amazonas Department. This difference may be due to the fact that from the 23 indigenous communities, only 8 have health stations (including the local hospital in Puerto Nariño, which is the only one capable of diagnosing TB by Ziehl-Neelsen staining). Given the geographical conditions, this population has very poor access to health services, taking several hours of commuting to get to health stations. In fact, most of the population assessed never had a medical consultation before this project, as they were treated by their own traditional medicine.

Among 74 clinical isolates recovered, 70 were obtained from patients who recognized themselves as indigenous. 98.6% of clinical isolates were susceptible to rifampicin and isoniazid, while 1.4% was MDR-TB. Polyclonal infections were detected in two patients (bringing the total number of genotypes to 76). However, a limitation in the study was the impossibility of taking new samples from these patients and of separating the colonies individually; nonetheless, cross-contamination could be ruled out since the samples from both patients were taken/treated on different days. The highest proportion of patients were men (61%), which is consistent with previous observations reported by the INS for 2015, showing that 63.3% of tuberculosis patients in Colombia were male [23]; and with the WHO global TB report showing that 56% of new cases in 2015 ocurred in men [24]. However, the high number of cases in children below 15 years (29/74 or 39.19%) is worth highlighting, which can be considered as sentinel events suggestive of recent transmission from sick adults, as well as indicative of a lack of control and proper treatment of primary cases. Although it is accepted that transmission in communities with high incidences of tuberculosis occurs through household contacts (i.e., by family members or close friends); transmission also occurs outside the home, yet inside the community [25]. The BCG immunization was determined by vaccination scar, with 47/74 (63.51%) of patients having a scar; while in the 1–14 age group, only 14/29 (48.28%) patients presented a scar (data not shown). Although there is no specific correlation with BCG immunization, studies have shown that vaccination reduces the risk of infection [26], and this could explain, in part, the higher number of cases in our pediatric patients.

This is the first study describing transmission of MTB using spoligotyping and 24-loci MIRU-VNTR in indigenous and non-indigenous population of the Colombian Amazon (Puerto Nariño). The phylogeographic studies carried out in general population of Colombia showed a predominance of the LAM and Haarlem sub-lineages, followed by T sub-lineages; for example, in a study in three cities from 2005 to 2008 in Colombia (Medellín, Cali and Popayán), Realpe et al. showed that 82.8% of circulating strains belonged to the LAM and Haarlem lineages [27]. Similarly, Cerezo et al. determined the population structure of *M. tuberculosis* in isolated clinical strains in Bogotá, finding a predominance of the LAM and Haarlem sub-lineages with 49.3% and 25% of the 152 clinical isolates studied, respectively [28]. The current study focusing

on the indigenous population of Puerto Nariño, Colombia, was partially consistent with previous reports with a predominance of the LAM sub-lineages (71.62% of the isolates); nonetheless, all the remaining isolates belonged to the T sublineage (27.03% of the isolates), while not a single Haarlem sub-lineage isolate was detected in the studied population. A possible explanation for this observation may be due to the low genomic diversity in the isolates obtained, since only 5 different spoligotype patterns were obtained, including 1 orphan pattern; while the study by Realpe et al. determined 84 spoligotype patterns, including 20 orphan patterns; and the study by Cerezo et al. showed 43 spoligototype patterns. The studies carried out in Brazil (that borders Colombia) showed similarities in the distribution of the sub-lineages. A study in 2006, carried out in indigenous population, showed that strains of *M. tuberculosis* circulating in the Suruí town, Brazil, belonged to the LAM genotype [29], while the study carried out in Minas Gerais, Brazil in 2004 showed that 55.3% of clinical isolates belonged to the LAM sublineage, 10.5% belonged to the T sublineage and 7% to the Haarlem sublineage, although it is important to emphasize that this study considered the general population and not just indigenous people [30].

Within the LAM sublineage, SIT95/LAM6 genotype was predominant with 50% of total clinical isolates belonging to this pattern in our study (Fig 2, Table 2). No report of this genotype was found previously in Colombia, however, this genotype was previously reported in 2012 from Peru (which borders Colombia) in 5/323 (1.55%) of pulmonary TB isolates in medical centers of Lima [31], and even at a lower rate of 0.25% in another study from Lima published in 2013 [32]. Similarly, SIT95/LAM6 genotype strains were found at low proportions (1.8%) in 2011 in the state of Minas Gerais, Brazil [30]. Furthermore, it is important to indicate that these studies were carried out in general population.

SIT42/LAM9 is the genotype with the highest circulation in general population of Colombia according to the studies carried out earlier with 29.9% [27] and 24.7% [33] of all isolates studied. In our study, this genotype represented only 2.7% of the isolates from indigenous population, underlining the differences among the strains circulating among indigenous vs. general population in Colombia. Finally, the SIT1355/LAM genotype (18.92% of strains in the present study) was already reported in a retrospective descriptive study covering the period 2009–2014 from the indigenous population in Colombia [34]. Out of 234 strains genotyped, 12 (5.1%) belonged to the Amazonas, and only 1/12 showed a genotype similar to SIT1355 found in the present study. Thus, the observation of SIT1355 strains in the present study is in agreement with earlier finding by Puerto et al. [34] in Ticuna ethnic group in Amazon. Additionally, SIT1355/LAM strains were reported in studies conducted in general population from Peru by Taype et al. [31] in 4.33%, and Sheen et al. [32] in 3.02% of the clinical isolates. Interestingly, SIT42/LAM9 which may be supposed to be an ancestor of SIT95 and SIT1355, was found only twice in our study. The visible scarcity of SIT42 in the studied region may suggest that evolution of SIT95 and SIT1355 (which only differ by one spacer) occurred more specifically in Puerto Nariño, where they became more predominant.

Within the T-sublineage, the SIT392/T2 genotype (27.03%) was reported, however, there is no report of this genotype in different studies conducted in Colombia (Fig 3). In Brazil, a study conducted in Porto Alegre found a clinical isolate (0.42%) belonging to SIT392/T2 although it is not specified whether this isolate corresponded to a patient belonging to an indigenous population [35]. The MSTs drawn in Fig 3 may indicate that specific genotypes are circulating among the indigenous population of Colombia (as these genotypes are particularly different from genotypes generally found in the country). LAM lineage seemed to be more present among indigenous people. Moreover, SIT392 was previously labeled as belonging to T family because of the absence of spoligotyping spacer 25 (which is usually present for patterns belonging to LAM lineage). However, one may notice that spacers 21–24 and 33–36, which

represent a specific signature of spoligotypes belonging to LAM lineage, are missing in SIT392 pattern. SIT392 pattern could be reclassified as belonging to LAM lineage. Further analyses would be needed to confirm this case.

It is noteworthy that clusters 4 and 5 share the same 24 loci MIRU-VNTR yet the discrimination is made by spoligotyping (SIT95/LAM6 and SIT392/T2, respectively). Note that SIT392/T2 differs from SIT95/LAM6 by absence of a block of 4 adjacent spacers 25–28 (Table 2). At this point, we have no explanation other than the temporal stability of the analyzed targets. Savine et al. demonstrated that 12 loci MIRU-VNTRs are stable for up to 6 years (MIRUs: 2, 4, 10, 16, 20, 23, 24, 26, 27, 31, 39 and 40) [36]. On the other hand, Niemann, Richter, & Rüsch-Gerdes show that the stability of the DR region is at least 772 days [37]. This change in the spoligotype could be interpreted as stochastic, since the DR region is a hotspot region for the integration of IS*6110* [38]; leading to unamplified spacer(s) within the DR locus of *M. tuberculosis* [39, 40]; a fact that should be investigated in future studies by studying the influence of the IS*6110* preferential insertion sites within the DR locus of these cluster isolates.

The typing by 24-loci MIRU-VNTRs allowed the identification of polyclonal infections in 2 patients cohabiting in the same community (Atacuari), as affirmed by the presence of two bands in 3 different MIRU loci. Several studies have demonstrated the clinical utility of the MIRU-VNTR technique for routine molecular typing of the *M. tuberculosis* isolates, as well as for detecting cases of polyclonal infections–particularly helpful for adjusting clinical treatments due to the heterogeneity that the mycobacterial population can present within the host [41, 42]. In addition, molecular typing allowed the identification of 14 different genotypes, 6 forming clusters and 8 corresponding to unique patterns. However, with 19 different genotypes of which 11 corresponded to unique patterns, the combination of both spoligotyping and MIRU-VNTR was undoubtedly more discriminative. Nevertheless, the overall lower genomic diversity observed in our study suggests that clones of *M. tuberculosis* circulating in indigenous population in Puerto Nariño have been maintained over time, considering that 7 loci MIRU-VNTR (26, QUB-26, 40, 1955, 42, 46 and 49) were polymorphics and the difference in numbers of copies was low, which is consistent with stepwise mutation mechanisms, i.e. sequential addition or deletion of repetitive units [36]. A valid strategy to have a greater discrimination power is to use a larger set of 24 loci for genotyping. On the other hand, the social organization based on marriage between exogamic alliances (tribes) within the Ticuna people, causing the interaction of its inhabitants just among them, may be another cause of this observed genomic homogeneity [43].

It is interesting that all clinical isolates present a signature compatible with the LAM sublineage RD115 (i.e. MIRU 2: 1 repeat) [44]. Although most of the strains circulating in Colombia belong to the RD-Rio sub-lineage, it is possible that few strains were established in the population, considering that the population of Puerto Nariño is relatively new, with the first settlers registered in 1940 (foundation: 1961) [45].

Regarding the comparison with the MIRU-VNTR patterns found in other studies, Cerezo et al. found MIRU International Type (MIT) 190, with a scheme of 12-loci MIRU-VNTRs, in 7.2% clinical isolates [28]. In the same way, in Peru, Taype et al., [31] found MIT 190 in 2.5% of clinical isolates, plus MIT 201 in 7.1%. In the area addressed, MIT 190 was found in 2/76 (2.6%) clinical isolates and MIT 201 in 36/76 (47.4%). As can be seen, MIT 190 has low circulation both in the areas addressed in the cited studies and Puerto Nariño. On the other hand, MIT 201 is, together with MIT 328, the predominant MIT (47% and 46%, respectively). One could speculate on a founder effector, since between both MITs, there is only a allelic change (MIRU 26), although this is based on a scheme of 12 loci. It cannot be fully compared as previous works did not use the 24 loci scheme; furthermore, the patterns obtained are orphans with the 24 loci scheme.

From contact tracing (n = 62), 12 additional cases were identified, of which 6 (50%) presented the same genotype as their respective index cases, this being a high percentage, according to what was reported by Buu et al., and Verver et al., who found 17% and 19% of contacts, respectively, with the same genotype as their index cases [46, 47]; which suggests a high home transmission; although transmission outside the home cannot be ruled out since the genotypes are also circulating in the community. The number of contacts with TB found demonstrates the importance of active search, especially in settings with high incidence [48].

In total, 63/74 clinical isolates were grouped in 8 clusters and although epidemiological links were established for only 10/63 (15.87%) patients, it is not possible to rule out greater transmission events for remaining patients since they frequently shared same space/time, though we could not establish a precise time of interaction. The use of classical genotyping methods and the restricted number of isolates may represent a limit in this study. Further studies using whole genome sequencing (WGS) or single nucleotide polymorphism (SNP) could be conducted in the future in order to better identify MTBC strains circulating among specific populations of Colombia. Additional investigations should also be performed within the non-indigenous population of Puerto Nariño, in order to better assess MTBC strains circulating in the region.

In conclusion, the present study indicates a chain of potentially active transmission in indigenous population, with a low genomic diversity of circulating strains in Puerto Nariño-Amazonas, Colombia. The results obtained also showed that specific genotypes circulating among the Indigenous population of Puerto Nariño (Amazonas) are significantly different from genotypes generally found in the country, according to the available studies. The genotyping of circulating strains in specific populations is a strategy for the improvement of TB control programs, e.g. by confirming epidemiological links or estimating cases of active transmission (with the same genotype). The percentage of clustered strains is a performance indicator of the TB control program that should be reduced as an effect of early detection and treatment of the disease. In addition, the study shows the need to increase efforts to detect and control tuberculosis in vulnerable populations such as indigenous population.

## Supporting information

**S1 Data.**
(XLS)

**S1 Fig. Puerto Nariño.** TB cases are indicated according to the genotypes found. Light blue: cluster 3; green: cluster 4; purple: cluster 5; blue: cluster 6; brown: orphan patterns. Scale bars indicate distance in meters. Reprinted from OpenStreetMap and QGIS 2.18.9 Las Palmas under a CC BY-SA 2.0 license.
(PDF)

**S2 Fig. 20 de Julio.** TB cases are indicated according to the genotypes found. Purple: cluster 5; blue: cluster 6. Scale bars indicate distance in meters. Reprinted from OpenStreetMap and QGIS 2.18.9 Las Palmas under a CC BY-SA 2.0 license.
(PDF)

**S3 Fig. Puerto esperanza.** TB cases are indicated according to the genotypes found. Red: cluster 1; light blue: cluster 3; brown: orphan pattern. Scale bars indicate distance in meters. Reprinted from OpenStreetMap and QGIS 2.18.9 Las Palmas under a CC BY-SA 2.0 license.
(PDF)

**S4 Fig. San Juan del Soco.** TB cases are indicated according to the genotypes found. Light blue: cluster 3; green: cluster 4; brown: orphan patterns. Scale bars indicate distance in meters. Reprinted from OpenStreetMap and QGIS 2.18.9 Las Palmas under a CC BY-SA 2.0 license. (PDF)

**S5 Fig. Tipisca.** TB cases are indicated according to the genotypes found. Purple: cluster 5; brown: orphan pattern. Scale bars indicate distance in meters. Reprinted from OpenStreetMap and QGIS 2.18.9 Las Palmas under a CC BY-SA 2.0 license. (PDF)

**S6 Fig. 12 de Octubre.** TB cases are indicated according to the genotypes found. Light blue: cluster 3. Scale bars indicate distance in meters. Reprinted from OpenStreetMap and QGIS 2.18.9 Las Palmas under a CC BY-SA 2.0 license. (PDF)

**S7 Fig. 7 de Agosto.** TB cases are indicated according to the genotypes found. Red: cluster 1; yellow: cluster 2; green: cluster 4; brown: orphan pattern. Scale bars indicate distance in meters. Reprinted from OpenStreetMap and QGIS 2.18.9 Las Palmas under a CC BY-SA 2.0 license. (PDF)

**S8 Fig. Boyahuazu.** TB cases are indicated according to the genotypes found. Red: cluster 1; light blue: cluster 3; green: cluster 4; brown: orphan pattern. Scale bars indicate distance in meters. Reprinted from OpenStreetMap and QGIS 2.18.9 Las Palmas under a CC BY-SA 2.0 license. (PDF)

**S9 Fig. Ticoya.** TB cases are indicated according to the genotypes found. Red: cluster 1; green: cluster 4; purple: cluster 5; blue: cluster 6. Scale bars indicate distance in meters. Reprinted from OpenStreetMap and QGIS 2.18.9 Las Palmas under a CC BY-SA 2.0 license. (PDF)

**S10 Fig. San Francisco.** TB cases are indicated according to the genotypes found. Yellow: cluster 2; light blue: cluster 3; green: cluster 4; purple: cluster 5. Scale bars indicate distance in meters. Reprinted from OpenStreetMap and QGIS 2.18.9 Las Palmas under a CC BY-SA 2.0 license. (PDF)

**S11 Fig. Naranjales.** Naranjales community (a) also includes a small territory named Santa Clara (b), approximately 1 km away. TB cases are indicated according to the genotypes found. Light blue: cluster 3; green: cluster 4; purple: cluster 5. Scale bars indicate distance in meters. Reprinted from OpenStreetMap and QGIS 2.18.9 Las Palmas under a CC BY-SA 2.0 license. (PDF)

**S12 Fig. Spoligoforest tree drawn as a hierarchical layout, based on all Euro-American spoligotypes detected in Colombia (n = 938, including the 74 isolates from the present study that comprised 76 genotypes).** Each spoligotype pattern from the study is represented by a node with area size being proportional to the total number of isolates with that specific pattern (number shown in brackets under the SIT number). Changes (loss of spacers) are represented by directed edges between nodes, with the arrowheads pointing to descendant spoligotypes. Nodes corresponding to spoligotypes from this study (focusing on Indigenous population) were colored in green. (PDF)

## Acknowledgments

We thank Thierry Zozio for his help with SITVIT2 database query.

## Author Contributions

**Conceptualization:** Martha Isabel Murcia.

**Data curation:** Alejandro Vega Marín.

**Formal analysis:** Alejandro Vega Marín, Nalin Rastogi, David Couvin, Viviana Mape.

**Funding acquisition:** Martha Isabel Murcia.

**Investigation:** Alejandro Vega Marín, Martha Isabel Murcia.

**Methodology:** Alejandro Vega Marín.

**Project administration:** Martha Isabel Murcia.

**Supervision:** Martha Isabel Murcia.

**Writing – original draft:** Alejandro Vega Marín, Martha Isabel Murcia.

**Writing – review & editing:** Alejandro Vega Marín, Nalin Rastogi, David Couvin, Martha Isabel Murcia.

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
