## [Decision Letter · Decision Letter 0]

16 Oct 2020

PONE-D-20-25842

First approach to the population structure of Mycobacterium tuberculosis complex in the indigenous population in Puerto Nariño-Amazonas, Colombia

PLOS ONE

Dear Dr. Murcia,

Thank you for submitting your manuscript to PLOS ONE. After careful consideration, we feel that it has merit but does not fully meet PLOS ONE’s publication criteria as it currently stands. Therefore, we invite you to submit a revised version of the manuscript that addresses the points raised during the review process.

While the reviewers were overall positive they made a number of comments. In addition I read your manuscript in its entirety and my comments also follow this letter.

We look forward to receiving your revised manuscript.

Kind regards,

Igor Mokrousov, Ph.D., D.Sc.

Academic Editor

PLOS ONE

Journal Requirements:

3.We note that [Figure(s) S1-11] in your submission contain [map/satellite] images which may be copyrighted. All PLOS content is published under the Creative Commons Attribution License (CC BY 4.0), which means that the manuscript, images, and Supporting Information files will be freely available online, and any third party is permitted to access, download, copy, distribute, and use these materials in any way, even commercially, with proper attribution. For these reasons, we cannot publish previously copyrighted maps or satellite images created using proprietary data, such as Google software (Google Maps, Street View, and Earth). For more information, see our copyright guidelines: http://journals.plos.org/plosone/s/licenses-and-copyright.

1.    You may seek permission from the original copyright holder of Figure(s) [S1-11] to publish the content specifically under the CC BY 4.0 license. 

Additional Editor Comments (if provided):

1. Major concern. I would suggest that you do NOT build tree based on MIRU and spoligo together.Rather it makes more sense to reanalyse your data in terms of clade assignment based only on 24-MIRU loci analysed together with 186 profiles in MIRU-VNTRplus. Otherwise addition of spoligo may create a confusion.When i did this analysis (with your file you had sent me by email), it is clear that ALL your isolates belong to LAM and its RD115 sublineage (signature MIRU10 — 1 repeat unit — see https://pubmed.ncbi.nlm.nih.gov/27001605/). And this is unusual and interesting as half of LAM in Colombia belong to RD-Rio.Please see attachment file miruvntr.pdf of the dendrogram where all your isolates are located well within LAM reference profiles. I realise that it may be a challenge but there is a simple PCR-RFLP to detect LAM specific SNP in Rv0129c and it would be most advisable to run this test on your isolates.

2. Please provide excel file with all data

3. All figures (original files) are of low resolution.

Reviewers' comments:

Reviewer's Responses to Questions

**Comments to the Author**

1. Is the manuscript technically sound, and do the data support the conclusions?

Reviewer #1: Yes

Reviewer #2: Yes

2. Has the statistical analysis been performed appropriately and rigorously? 

Reviewer #1: Yes

Reviewer #2: N/A

3. Have the authors made all data underlying the findings in their manuscript fully available?

Reviewer #1: Yes

Reviewer #2: Yes

4. Is the manuscript presented in an intelligible fashion and written in standard English?

Reviewer #1: Yes

Reviewer #2: Yes

5. Review Comments to the Author

Reviewer #1: 

1.It may be worth reconsidering the relevance of Figure 3. A tree based only on spoligotyping data (single DR locus with non-independent characters/spacers) can lead to false clustering.

2. An interesting and important fact is that almost half of the samples were children. Perhaps it is worth clarifying is it a trend for this region? Is BCG vaccine given to children in this region?

Reviewer #2: Minor comments:

The manuscript presents a complete study of the tuberculosis situation in an indigenous population of a region of Colombia´s Amazon, Puerto Nariño-Amazonas. The study is based on molecular epidemiology using genotyping by Spoligotyping and MIRU-VNTRtyping. The molecular information obtained is completed with epidemiological and geographic data that allow a good picture of the transmission of tuberculosis in the indigenous population studied. In addition, a comparison with the genotypes of the circulating strains in non-indigenous areas of Colombia and in surrounding countries is integrated into the study. I find it a very interesting study, well done and clearly exposed, so I recommend its publication.

A detailed reading of the work has generated some questions and comments, which I hope can help clarify some aspects and enrich the discussion:

1.- In the description of the study sample, it is mentioned that 12 of the studied patients were identified by contact tracing. Was the genotype of these patients the same of their original contacts? I mean, was the strain of the patient identified by trace matching the strain of the patient that alerted that contact? If this were not the case, it would be warning of a potentially greater number of infected patients than those studied. It is an indirect measure of a high community transmission and of the effectiveness of screening; perhaps suggesting the need to extend screening to an unsuspected population.

2.- In the Figure 1 caption (dendrogram) there is an error in the description of the fields included, the fourth field starting from the left is the MIRU-VNTR pattern and not SITs based on SpolDB4, isn´t?

3.- In line 126 on page 6 when referring to DNA extraction, it should be indicated that it was extracted from the grown culture, because as it is written it could be confused with direct extraction from a clinical sample, and I understand that it is not the case. Perhaps this can be indicated in the previous section “DNA extraction”.

4.- In the population study, it is commented that 25.7% of the TB patients belonged to the Yagua ethnic group, but this group only accounts for 7.4% of the total population, it is, therefore, the ethnic group with the greatest affectation. What reason could there be for this situation? It does not seem linked to the communities where they reside, since this ethnic group is not particularly represented in the 2 populations with the highest number of tuberculosis cases, Puerto Nariño and San Francisco. Is there an epidemiological explanation or of any other kind that can explain it? It is very interesting and curious. Is there anything described in relation to this ethnic group that can justify this high incidence? Could they be, perhaps, the most exposed to a non-indigenous population?

5.- With respect to patients with polyclonal/mixed infection:

a) How was the genotype of both strains involved deduced? Through the gel, it can only be interpreted that there is a locus with 2 alleles, but not which allele belongs to which strain. Were colonies picked to separate variants and genotyped to assign corresponding alleles? Otherwise, the allele combination may be different from that proposed in the manuscript and the genotypes, therefore, different. Was there a difference in intensity in the allele bands that allowed you to deduce which alleles were shared by the same strain? If one of the strains were represented in a lower proportion than the other, it is possible that the genotype can be deduced based on the intensities of the gel bands.

b) Has cross-contamination in the lab been ruled out? It is very strange that two patients have a mixed infection with the same 2 strains. Analysing the processing date of both strains, this laboratory contamination could be ruled out since if they had been processed at different times they could not have been mixed in the laboratory.

6) The paragraph explaining the cluster distribution in Figure 1 is confusing (199-206).

I understand that the final 8 clusters referred to are those named as 1-6 plus clusters a and b. If this is the case, in the last sentence, the "Additionally" (line 205) is left over, since they have already been included in the previous paragraph, it gives the feeling that they are 8 + 2. If I am wrong in my deduction, then it should be explained in more detail which 8 clusters the authors refer to, because it is not clear.

7.- For the 6 cases of indigenous people that were identified as relapses (lines 272-273), was it possible to analyze whether they were reactivations or reinfections with a different strain? Were the strains from the first episodes available for genotyping by MIRU-VNTR? Due to the high rates of tuberculosis in this population, new exposure and therefore reinfection cannot be ruled out. It would be very interesting if you had made him comment on it.

8.-Regarding comparisons with other circulating strains, I´m lacking some comparisons mediated by MIRU-VNTR genotypes.

a) Have the study MIRU-VNTR genotypes been compared with those previously published by Cerezo and Realpe? Or with any other previous study that presents genotypes by MIRU-VNTR of circulating strains in Colombia or neighbouring countries? It is possible that some genotype was previously circulating and that once it was introduced into the indigenous population, it established itself as the dominant strain and subsequently evolved there endemically.

b) Did the strains with the LAM6-95 genotype found in Peru and Brazil (lines 327-334) share also the MIRU-VNTRtype (1355) with the study strains?

It is strange as well as very interesting that the spoligotype breaks the genotypes formed by MIRU-VNTR when it is usually the contrary. It is true that the study population is very special, and this probably has something to do with it. I ask the authors as experts (out of curiosity) if they think it is likely that there is homoplasy between MIRU-VNTR genotypes, could it be?

6. PLOS authors have the option to publish the peer review history of their article (what does this mean?). If published, this will include your full peer review and any attached files.

Reviewer #1: No

Reviewer #2: No

---

## [Author Response · Author response to Decision Letter 0]

18 Dec 2020

We would like to thank you as well as the reviewers for a constructive criticism of our paper. As you will see in the attached file, we have thoroughly revised the paper following a careful review of comments/observations made; the changes made are detailed in order to meet the scientific and editorial needs underlined.

---

## [Editor Report · Decision Letter 1]

22 Dec 2020

First approach to the population structure of Mycobacterium tuberculosis complex in the indigenous population in Puerto Nariño-Amazonas, Colombia

PONE-D-20-25842R1

Dear Dr. Murcia,

I am pleased to inform you that your manuscript has been judged scientifically suitable for publication and will be formally accepted for publication once it meets all outstanding technical requirements.

Kind regards,

Igor Mokrousov, Ph.D., D.Sc.

Academic Editor

PLOS ONE
---

## [Editor Report · Acceptance letter]

28 Dec 2020

PONE-D-20-25842R1 

First approach to the population structure of *Mycobacterium tuberculosis* complex in the indigenous population in Puerto Nariño-Amazonas, Colombia 

Dear Dr. Murcia:

I'm pleased to inform you that your manuscript has been deemed suitable for publication in PLOS ONE. Congratulations! Your manuscript is now with our production department. 

Kind regards, 

on behalf of

Dr Igor Mokrousov 

Academic Editor

PLOS ONE